

# Latent Linear Adjustment Autoencoders v1.0: A novel method for estimating and emulating dynamic precipitation at high resolution

Christina Heinze-Deml[1], Sebastian Sippel[1,2], Angeline G. Pendergrass[3,2], Flavio Lehner[2,3], and
Nicolai Meinshausen[1]

[1]Seminar for Statistics, ETH Zurich, Zurich, Switzerland
[2]Institute for Atmospheric and Climate Science, ETH Zurich, Zurich, Switzerland
[3]Climate and Global Dynamics Laboratory, National Center for Atmospheric Research, Boulder, Colorado

**Correspondence:** Christina Heinze-Deml (heinzedeml@stat.math.ethz.ch)

**Abstract.** A key challenge in climate science is to quantify the forced response in impact-relevant variables such as precipitation against the background of internal variability, both in models and observations. Dynamical adjustment techniques aim to remove unforced variability from a target variable by identifying patterns associated with circulation, thus effectively acting as a filter for dynamically-induced variability. The forced contributions are interpreted as the variation that is unexplained

by circulation. However, dynamical adjustment of precipitation at local scales remains challenging because of large natural variability and the complex, nonlinear relationship between precipitation and circulation particularly in heterogeneous terrain. Building on variational autoencoders, we introduce a novel statistical model—the Latent Linear Adjustment Autoencoder—that enables estimation of the contribution of a coarse-scale atmospheric circulation proxy to daily precipitation at high-resolution and in a spatially coherent manner. To predict circulation-induced precipitation, the Latent Linear Adjustment Autoencoder

combines a linear component, which models the relationship between circulation and the latent space of an autocoder, with the autoencoder's nonlinear decoder. The combination is achieved by imposing an additional penalty in the cost function that encourages linearity between the circulation field and the autoencoder's latent space, hence leveraging robustness advantages of linear models as well as the flexibility of deep neural networks. We show that our model predicts realistic daily winter precipitation fields at high resolution based on a 50-member ensemble of the Canadian Regional Climate Model at 12-km

resolution over Europe, capturing for instance key orographic features and geographical gradients. Using the Latent Linear Adjustment Autoencoder to remove the dynamic component of precipitation variability, forced thermodynamic components are expected to remains in the residual, which enables the uncovering of forced precipitation patterns of change from just a few ensemble members. We extend this to quantify the forced pattern of change conditional on specific circulation regimes. In addition, we briefly illustrate one of multiple possible further applications of the method: a weather generator that emulates

climate model simulations of regional precipitation at high resolution by bootstrapping circulation patterns. Other potential applications include addressing detection&attribution at sub-continental scales, statistical downscaling and transfer learning between models and observations to exploit the typically much larger sample size in models compared to observations.



## 1 Introduction

Precipitation is a key climate variable that is highly relevant for impacts such as floods or meteorological drought. Precipitation
simulations at high-resolution (e.g., Prein et al., 2017) are required for adaptation planning for local and regional precipitation
change in a warming climate. However, precipitation shows large natural variability (Deser et al., 2012), and its relationship
with atmospheric circulation is complex and non-linear, in particular at local to regional scales and in heterogeneous terrain
(e.g., Zorita et al., 1995). Moreover, projected changes in precipitation are unevenly distributed across the distribution of
precipitation intensity (Allen and Ingram, 2002; Held and Soden, 2006; Pendergrass, 2018). Scaling rates depend on the return
period, region, temperature and moisture availability (Prein et al., 2017), and changes in circulation during precipitation events
(Shepherd, 2014; Fereday et al., 2018). Hence, it is a key challenge to identify, understand and interpret patterns of forced
precipitation change in model simulations and observations.

Dynamical adjustment techniques have been developed to separate forced and internal variability via a co-interpretation of
target variables such as temperature or precipitation using circulation information: A circulation proxy (such as a sea level
pressure pattern) is used to estimate the circulation-induced (dynamic) contribution to temperature or precipitation variability.
For example, dynamical adjustment of precipitation has revealed that the spatial pattern and amplitude of observed residual
(thermodynamic) precipitation trends at the scale of the entire Northern hemisphere mid- and high-latitude land area are in
good agreement with the expected anthropogenically-forced trends from model simulations (Guo et al., 2019). Similarly, in
Europe, Fereday et al. (2018) showed that thermodynamic forced changes in future winter precipitation are in relatively good
agreement among models, while large uncertainties remain in simulated forced circulation changes that may affect precipi-
tation. While internal and forced components of precipitation variability and change can be decomposed in large ensembles
of model simulations (Deser et al., 2012; von Trentini et al., 2019; Leduc et al., 2019), high resolution large ensembles are
prohibitively expensive. It would be beneficial to be able to estimate and identify forced precipitation patterns from only a few
ensemble members at impact-relevant regional spatial scales.

Techniques for dynamical adjustment have relied largely on linear regression (Wallace et al., 1995, 2012; Smoliak et al.,
2015; Sippel et al., 2019) or circulation analogue techniques (Yiou et al., 2007; Deser et al., 2016). Because the fraction of
variability that can be explained by these techniques is limited, dynamical adjustment has so far been applied on large spatial
and temporal scales (e.g., Guo et al., 2019), and has also been more successful for temperature trends than for precipitation.
Applying it to precipitation on local or regional scales at high-resolution remains a challenge.

Beyond techniques of dynamical adjustment, large-scale simulated patterns of climate variables have been used to emulate
climate model simulations in order to generate larger samples of realizations of the climate system (combining dynamical
internal variability and the forced response) for purposes like impact modeling (Link et al., 2019; Beusch et al., 2020; Alexeeff
et al., 2018). While these approaches have been used successfully for emulating large-scale temperatures, emulating precip-
itation at local or regional scales at high temporal frequency remains challenging due to nonlinearity and the complexity of
processes involved (Link et al., 2019; Beusch et al., 2020).





In this work, we leverage recent advances in machine learning to propose a novel statistical model—the Latent Linear Adjustment Autoencoder—suitable for dynamical adjustment and emulating dynamically-induced variability in daily, high-resolution precipitation fields. In recent years, deep learning techniques have gained in popularity in machine learning due to large improvements in neural network architectures, optimization algorithms as well as computing power and frameworks.

Among the class of deep generative models, the introduction of variational autoencoders (VAEs) (Kingma and Welling, 2014; Rezende et al., 2014) started a whole subfield in deep learning research. Among other things, the popularity of VAEs is due to their ability to generate realistic new images, in addition to being a powerful nonlinear dimensionality reduction technique. Our proposed Latent Linear Adjustment Autoencoder extends the standard VAE model appropriately to enable the climate applications of interest.

Specifically, during training the Latent Linear Adjustment Autoencoder encodes daily precipitation fields into a low-dimensional latent space and subsequently decodes them for reconstruction. In addition, we formulate the objective function such that the latent space can be regressed linearly on the circulation proxy. For dynamical adjustment, we use the estimate of the latent space based on circulation, which is then decoded for predicting daily precipitation fields at high spatial resolution. In other words, the final model is nonlinear, consisting of a linear part and a nonlinear part, where the latter is a deep neural network. It enables prediction of the portion of the precipitation field that can be explained by circulation (i.e., the dynamic component of precipitation). Beyond the dynamical adjustment application, the model can also be used to emulate dynamically-induced variability in daily precipitation fields by bootstrapping the circulation patterns.

In summary, the objectives of this paper are the following:

1. Introduce a novel statistical model—the Latent Linear Adjustment Autoencoder—as a versatile technique for applications in climate science, particularly for making better use of high-resolution climate simulations by estimating circulation-induced (dynamic) precipitation at high resolution from coarse-scale circulation information.

2. Illustrate the Latent Linear Adjustment Autoencoder by applying it to dynamical adjustment of daily high-resolution precipitation from simulations over Central Europe.

3. Illustrate that our technique can be extended to emulate the dynamic component of daily precipitation fields, akin to weather generators.

## 2 Dynamical adjustment using statistical learning

Following Smoliak et al. (2015) and Sippel et al. (2019), we frame dynamical adjustment as a statistical learning problem. Let $Y \in \mathbb{R}^{h \times w}$ be the climate variable of interest on a spatial field of size $h \times w$ and let $X \in \mathbb{R}^p$ be input features. In the following, we consider daily precipitation fields for $Y$ and empirical orthogonal function (EOF) time series of sea-level pressure (SLP) for $X$ as a proxy for circulation. The EOF time series are detrended (as described below) and scaled to unit-variance; in the EOF computation, we do not weight by area. A variety of climate variables instead of precipitation could be taken as $Y$; results for daily temperature can be found in Appendix B.





Let $\mathbf{X}$ be the $n \times p$ matrix where each column contains one input feature with $n$ "observations". Each "observation" is in our case a simulation from a Regional Climate Model (RCM), and each column corresponds to one of $p$ EOF components of SLP, from the RCM but at coarsened resolution. Let $\mathbf{Y}$ be the $n \times h \times w$ tensor that represents the precipitation intensity for each "observation" in a spatial field. Below, we present our proposed statistical model, which estimates the circulation-induced component of precipitation $\hat{\mathbf{Y}}_{\mathbf{X}}$,

$$\hat{\mathbf{Y}}_{\mathbf{X}} = f(\mathbf{X}), \tag{1}$$

where $f$ is a generic non-linear function. Let $\hat{\mathbf{R}}$ denote the residuals $\hat{\mathbf{R}} = \mathbf{Y} - \hat{\mathbf{Y}}_{\mathbf{X}}$ that remain. Since $\hat{\mathbf{Y}}_{\mathbf{X}}$ is the precipitation explained primarily by variations in circulation, $\hat{\mathbf{R}}$ is the precipitation primarily unexplained by circulation. If SLP is unaffected by external forcing, then this residual contains the signal induced by the thermodynamic component of the external forcing, since the variability due to circulation has been removed. If instead external forcing does affect SLP, then the dependence between $X$ and $Y$ that arises due to the common influence of the external forcing would bias the estimation of $f$. To avoid potential forced trends in SLP projecting onto the thermodynamic component of the external forcing in $Y$, we detrend the daily SLP EOF time series as follows. We ensure that they are orthogonal to the smoothed, first EOF of the January ensemble mean by regressing the time series against the ensemble mean, and using the corresponding residuals as input features $X$. The reasoning behind this step is that a forced trend in SLP will be approximately captured in the first EOF of the SLP ensemble mean. Here, we use the January ensemble mean as a proxy for December-February SLP. For simplicity, we refer to the detrended and scaled SLP EOF time series simply as the "SLP time series" in the following.

## 2.1 Latent Linear Adjustment Autoencoders: Proposed deep autoencoder model for dynamical adjustment

We build on variational autoencoders (VAEs; Kingma and Welling, 2014; Rezende et al., 2014), which can be understood as a (typically nonlinear) dimensionality reduction method. An autoencoder consists of an "encoder" $e$ that maps $Y$ to the low-dimensional latent space $L \in \mathbb{R}^l$, $L = e(Y)$, and a "decoder" $d$ which in turn maps $L$ to the reconstruction of $Y$, $\hat{Y} = d(L) = d(e(Y))$. This scheme is illustrated in Fig. 1, which depicts the reconstruction of precipitation fields. The VAE objective encourages the distribution of the latent space variables $L$ to be close to a chosen prior distribution, typically a standard multivariate Gaussian distribution, and also ensures that $\hat{Y} \approx Y$. The encoder and the decoder are parameterized as (deep) neural networks.

We extend the standard VAE model to make it suitable for dynamical adjustment by adding a linear component $h$ to the architecture. The linear component $h$ takes $X$ as input features and predicts the latent space variables $L$ of the VAE; thus we call the overall model the "Latent Linear Adjustment Autoencoder". Using an appropriate training objective (see Eq. (3)), we enforce that when linearly predicting the latent space variables $L$ with $X$, $\hat{L} = h(X)$, the resulting decoded prediction $\hat{Y}_X = d(\hat{L}) = d(h(X))$ to be close to $Y$. The motivation behind this loss function is that the combined model, which consists of the combination of $h$ and $d$, should explain as much variance in $Y$ as possible, while using only the input $X$. In other words, it should capture the circulation-induced signal in $Y$. The advantage of combining the linear model $h$ with the nonlinear decoder of the VAE $d$ is that the overall model is very expressive while the estimation of $h$ remains relatively simple.





In more detail, we consider the following objective to train the encoder $e$ and decoder $d$ with associated parameters $\theta = (\theta_e, \theta_d)$ of our proposed Latent Linear Adjustment Autoencoder:

$$\mathcal{L}_\theta = \mathcal{L}_{\text{VAE}} + \lambda \mathcal{L}_{\text{L}}. \tag{2}$$

$\mathcal{L}_{\text{VAE}}$ is the standard VAE objective for real-valued input data, consisting of a reconstruction loss and the Kullback–Leibler
divergence between the distribution of the encoded inputs and the prior distribution of the latent space, here chosen to be a standard multivariate Gaussian distribution (for details see Kingma and Welling, 2014; Rezende et al., 2014). $\mathcal{L}_{\text{L}}$ is the extension to the objective that we propose,

$$\mathcal{L}_{\text{L}} = \|\mathbf{Y} - \hat{\mathbf{Y}}_{\mathbf{X}}\|_2^2 = \|\mathbf{Y} - d(h(\mathbf{X}))\|_2^2, \tag{3}$$

and $\lambda$ is a tuning parameter that steers the relative importance of the two loss functions in the overall objective (Eq. (2)). The
130 autoencoder and the linear model are trained iteratively in an alternating fashion. In the first step, the objective $\mathcal{L}_\theta$ is optimized while $h$ is treated as fixed. In a second step, the linear component $h$ of the Latent Linear Adjustment Autoencoder is trained with squared error loss, treating the encoder and decoder parameters as fixed,

$$\mathcal{L}_{\theta_h} = \|\mathbf{L} - \hat{\mathbf{L}}\|_2^2 = \|e(\mathbf{Y}) - h(\mathbf{X})\|_2^2. \tag{4}$$

The parameters of the encoder and decoder $(\theta_e, \theta_d)$ and those of the linear model $h$, $\theta_h$, are coupled, since the linear model aims
to predict the latent space variables $L$, which are subject to change during the training of the encoder and the decoder. At the same time, the autoencoder should be trained *such that* a linear regression from $L$ on $X$ achieves a small error in Eq. (3), which is accomplished with this procedure for training the components. In practice, we train the model using the Adam optimizer (Kingma and Ba, 2015). All details related to training the model, such as architecture and hyperparameter choices, as well as code to reproduce our experimental results, can be found in Appendix A.

After training the components $e$, $d$ and $h$, we no longer need the encoder to perform dynamical adjustment on unseen test data. This is illustrated in Fig. 2. We predict the latent space variables with the linear model $h$, using the SLP time series as input $X$. The resulting predictions are fed to the decoder, which outputs predictions of the spatial field based only on $X$. In other words, we obtain the spatial field of precipitation which can be explained by circulation.

The spatial field is modelled jointly in our approach - the optimization is performed over the whole spatial field at once - in
contrast to Sippel et al. (2019), where a separate model needed to be trained for each grid cell. The joint modeling of the daily high-resolution precipitation field as a function of coarse-scale circulation also enables several additional climate applications which are discussed in Sect. 5.

## 3 Data and evaluation

### 3.1 Data

To evaluate our statistical model, we use the Canadian Regional Climate Model Large Ensemble (CRCM5-LE, (Leduc et al., 2019)) that is based on a dynamically downscaled version of the 50-member CanESM2 large ensemble (CanESM2-LE).





CanESM2-LE is an initial-condition ensemble of climate change projections (Kirchmeier-Young et al., 2017) run with the Canadian Earth System Model (Arora et al., 2011), globally on a spatial resolution of 2.8°. CanESM2-LE combines so-called macro and micro initializations: to achieve different 1950 ocean states (the macro initialization), five historical spinup runs are branched from a long pre-industrial (1850) control run, to which small random atmospheric perturbations and then time-varying forcings are applied until 1950. Then, for the micro ensemble (where members have the same initial conditions in the ocean but differ in their atmosphere), a second set of random perturbations is applied in 1950 to generate 10 runs spanning 1950-2006 for each of the 5 historical spinups, with time-varying historical forcing scenario applied through 2006. The runs continue from 2006 until 2100 with RCP8.5 forcing (Kirchmeier-Young et al., 2017).

This approach yields 50 approximately independent realizations of the climate system (Leduc et al., 2019). Each of the global simulations has been downscaled using the Canadian Regional Climate model version 5 (Martynov et al., 2013) to a resolution of approximately 0.11° ($\approx$ 12 km) over Europe for the 1950-2099 period, which yields the regional large ensemble CRCM5-LE. More details about the modeling setup and evaluation of the simulations are available in Leduc et al. (2019).

## 3.2   Experimental setup

### 3.2.1   Target variables

We focus on precipitation as the target climate variable throughout the main text. A subset of results for temperature can also be found in Appendix B. For the precipitation fields we apply a square-root transformation to stabilize the autoencoder training; the results presented in Sect. 4 are based on transforming the results back to the original scale ($\text{mm} \cdot \text{d}^{-1}$), except for some visualizations where indicated in the captions. The spatial fields we consider have 128x128 grid cells, i.e. they are a subset of the original 280x280 field, cropped around Central Europe (with boundaries 0-18.9°E and 42-54.8°N). We aggregate the data (which are hourly) to daily averages.

### 3.2.2   Input features

SLP is regridded to a spatial resolution of 1x1° before computing the EOFs as described in Sect. 2, so that the model predicts high-resolution precipitation from only a coarse-resolution proxy of atmospheric circulation. We aggregate the data (3-h) to daily averages. SLP data is also taken from the original 280x280, 0.12° resolution Euro-Cordex domain (WCRP, 2015) and regridded to a regular 1x1° grid that broadly covers the region of -15 to 35°E and 35-64°N (see Fig. 7a).

### 3.2.3   Time periods and training/test splits

We use RCM simulation data from 1955 to 2100 to allow for 5 years of spinup. We train our model using daily data from December-February from 9 ensemble members ("kba", "kbc", "kbe", "kbg", "kbi", "kbk", "kbm", "kbq", "kbs"). The training data comprises the years 1955-2070, the years 2071-2100 are used for testing. Furthermore, we evaluate our trained models on six ensemble members that were entirely left out of training ("kbb", "kct", "kcu", "kcv", "kcw", "kcx"). We refer to these as "holdout ensemble members".





### 3.2.4 Evaluation of predictions

To illustrate the spatial coherence of our approach, we show 5 example target (daily) "observations" of $Y$, their reconstructions $\hat{Y}$, and the predictions $\hat{Y}_X$ from the holdout ensemble member "kbb" (Fig. 3). To highlight the precipitation features, these examples are displayed after the data has been square-root-transformed. We compute the gridcell-wise mean squared error (MSE) of predictions $\hat{Y}_X$ and the proportion of explained variance ($R^2$) based on the daily data from the holdout ensemble member "kbb". We further evaluate the predictions within a dynamical adjustment framework described in the next paragraph.

### 3.2.5 Dynamical adjustment

We evaluate the extent to which the forced response of precipitation can be uncovered with a small number of ensemble members using dynamical adjustment (e.g., Deser et al., 2016). Specifically, we quantify how well the long-term "forced response" (i.e., the average across all 50 ensemble members) can be approximated by the residuals of our predictions (the difference between precipitation simulated by the RCM and the circulation-induced component of precipitation predicted by the Latent Linear Adjustment Autoencoder). Recall that we expect the residuals to contain thermodynamic component of change (Deser et al., 2016). In other words, dynamical adjustment acts as a filter for "circulation-induced" precipitation variability. In addition to analysing the forced response of mean precipitation, we evaluate the estimation of the forced precipitation response for two composites of atmospheric circulation based on EOF analysis of SLP.

## 4 Results

### 4.1 Reconstructed and predicted spatial fields

We begin by showing a selection of reconstructed precipitation fields $\hat{Y}$ from the holdout ensemble member "kbb" (center, Fig. 3), which illustrates the skill of the encoder and decoder, and predictions $\hat{Y}_X$ (right, Fig. 3), which illustrate the skill of the linear latent model $h$, against the original RCM-simultated precipitation $Y$ (left, Fig. 3). The reconstruction quality, i.e. the similarity between the left and the center column, is quite high, though not all fine details are reproduced (which is to be expected). The predictions $\hat{Y}_X$ (right column) are computed using the linear model $h$ and the decoder $d$ with SLP time series as inputs. For dynamical adjustment we use the residuals $\hat{R}$, which are computed as the difference between the original fields $Y$ (left column) and the predictions (right column).

The proposed model yields spatially coherent predictions and explains a large proportion of the variance of $Y$. Skill of precipitation predicted with the Latent Linear Adjustment Autoencoder is quantified in Fig. 4, which shows the gridcell-wise MSE over all December-February days for the holdout ensemble member "kbb". The spatial pattern of MSEs (Fig. 4) largely reflects the pattern of precipitation as simulated by the RCM. Prediction errors are high over heterogeneous terrain, likely linked to orographic precipitation, in particular on the western sides of mountain ranges such as the Alps in Central Europe, the Appenine in Italy, the Dinaric Alps in South-East Europe, and smaller ranges located in France and Germany. Prediction errors are also high at the west coast of the UK, whereas mean squared prediction errors appear relatively low over low altitude





regions (e.g., northern France, Benelux, and north Germany). The spatial pattern of $R^2$ (the fraction of explained variance,
Fig. 5) shows a more nuanced pattern dominated by a land-sea contrast. Over land, the circulation proxy generally explains a
high proportion of variance (up to $\approx 90\%$), especially on the western slopes of mountain ranges, which receive a large fraction
of their precipitation from large-scale circulation-induced events. In contrast, the fraction of variance explained by circulation-
induced precipitation is smaller on the eastern sides of mountain ranges, and particularly low over oceanic regions (which we
do not interpret in this study).

## 4.2   Extraction of forced precipitation trends at high spatial resolution

In this subsection, we evaluate our predictions of the circulation-induced component in the framework of dynamical adjustment
(Deser et al., 2016). That is, we test the extent to which the forced response of regional high resolution precipitation obtained
from averaging across the full 50-member ensemble can be approximated by the residuals of our predictions from a single or
from relatively few ensemble members. The thermodynamic component of the variation in precipitation, which is driven by
temperature change and unrelated to dynamically-induced variability, should remain in the residuals (e.g., Deser et al., 2016).
Dynamical adjustment, hence, acts to reduce short-timescale circulation-induced variability, thus increasing signal-to-noise
ratios of the long-term, forced thermodynamic component (Deser et al., 2016).

The effect of dynamical adjustment can be seen in Fig. 6. It shows time series of domain-average (land only) December-
February seasonal mean precipitation simulated by the high resolution RCM, the predictions of the circulation-induced com-
ponent for three holdout ensemble members, and the forced response. All three RCM ensemble members show an increasing
trend in seasonal mean precipitation across the 21st century, over which large inter-annual variability is superimposed. In con-
trast, the predicted circulation-induced components capture the inter-annual precipitation variability well, but they do not show
discernible trends. Consequently, the residuals have relatively smoothly increasing trends, which match the magnitude of the
forced precipitation trend well (Fig. 6, right panels). This demonstrates successful dynamical adjustment of continental-scale,
seasonal-mean precipitation using the Latent Linear Adjustment Autoencoder.

It is more challenging, however, to identify and evaluate the forced precipitation response at the local scale of individual
grid points. To this end, Fig. 7 shows the spatial pattern of forced 50-year (2020-2069) precipitation trends, the pattern of
dynamically adjusted precipitation trends, and "raw" precipitation trends in RCM simulations for three holdout ensemble
members. The forced response of winter precipitation change is dominated by a north-south contrast. The northern part of the
domain is projected to experience a precipitation increase in the 21st century, while decreases in precipitation are projected for
the southernmost part of the domain (mainly over the Mediterranean Sea). Increases in winter precipitation across most parts
of the domain are largely due to thermodynamic and lapse rate changes (e.g., Brogli et al., 2019). Locally, forced precipitation
trends are larger over heterogeneous terrain, which may be due to forced dynamic components that are independent of SLP
(Shi and Durran, 2014). The latter paper shows idealized simulations of the forced response of orographic precipitation, which
is dynamic, but it's driven by changes in vertical velocity on the upslope side that enhances orographic precipitation, which
could be separate from the changes in upslope wind speed and thus SLP. Vertical velocity and could increase because of the
increasing moisture with warming.



However, large variability in individual ensemble members is superimposed on the signal of forced change (Fig. 7, right), consistent with the large role of internal variability even on multi-decadal time scales (Leduc et al., 2019). For example,

ensemble member "kct" (second row) produces a relatively strong drying trend over northern Italy, which is entirely due to internal variability. The dynamically adjusted version of "kct" shows only very weak drying in northeast Italy, whereas it shows an increased precipitation trend consistent with the forced response at all other locations in northern Italy . Similar differences can be seen between the adjusted and unadjusted trends for the other holdout ensemble members. Overall, the correlation of 50-year trends at a single location from a single ensemble member with the trend in the 50-member forced response is rather

low (R = 0.4, RMSE = 0.098). However, the dynamically adjusted single ensemble members are more strongly correlated with the forced response and capture it more accurately (R = 0.58, RMSE = 0.052). Hence, to achieve the same RMSE of a single dynamically adjusted ensemble member, an ensemble of about four to six members would be required (Fig. 8).

### 4.3 Circulation regime specific forced precipitation trends at high spatial resolution

While dynamical adjustment of long-term trends of temperature and precipitation has become a standard tool for the detection

and forced thermodynamic trends (Smoliak et al., 2015; Deser et al., 2016; Guo et al., 2019; Lehner et al., 2018), a bigger challenge is to assess forced trends in specific circulation regimes. One example would be summer heat waves related to specific circulation conditions (Jézéquel et al., 2018).

Thus, we assess to what extent the forced precipitation response can be uncovered under specific circulation conditions from a small number of ensemble members. We create composites of the dominant mode of atmospheric winter circulation

over Europe as diagnosed by EOF analysis over the historical period (1955-2020) in the RCM simulations. The first EOF of the coarse-resolution SLP field is shown in Fig. 9 (top). The dominant mode has a meridional gradient, with low pressure anomalies over northern Europe and high pressure anomalies over the Mediterranean. Although the domain includes only a small fraction of the North Atlantic, the dipole character of the EOF spatial pattern resembles the North Atlantic Oscillation.

We now generate composites of 'EOF1+' and 'EOF1-' regimes by isolating days that exceed the 75th percentile ('EOF1+')

and those that fall below the 25th percentile ('EOF1-') in terms of the first principal component (Fig. 9, bottom). Note that the principal component time series associated with EOF1 does not show any discernible trend until the late 21st century, and only a very minor change thereafter, so we do not expect large forced changes in the SLP variability patterns over Europe. On winter days with strong positive EOF1 ('EOF1+', roughly analogous to NAO+), i.e. a pronounced north-south pressure gradient, increased westerly winds bring mild and moist air from the Atlantic into Central Europe (Fig. 9, bottom left). Conversely,

the opposite regime suppresses westerlies, hence inducing drier conditions on average (Fig. 9, bottom right) which are also accompanied by colder temperatures.

For the 50-year forced precipitation trend on 'EOF1+' winter days (obtained by averaging across all ensemble members), there is a more pronounced precipitation increase on the western slopes of the Alps and in most parts of the domain north and west of the Alps (Fig. 10, left panel). Meanwhile, precipitation decreases in the EOF1+ regime over southern Europe.

Raw simulations for sets of three holdout members show variable 50-year (2020-2069) precipitation trends under the 'EOF1+' regime (Fig. 10, right panel), where the spatial pattern of each set of three averaged members only weakly resembles





the forced pattern (pattern RMSE = 0.174). The dynamically adjusted holdout members reveal a pattern that more closely resembles the forced response pattern (Fig. 10, middle panel), where forced changes in mountainous regions are particularly well captured; the pattern RMSE is reduced substantially (RMSE = 0.085).

Forced precipitation trends for 2020-2069 under 'EOF1-' conditions differ from 'EOF1+' conditions due to a change in the synoptic situation: the forced spatial pattern has generally weaker precipitation changes (due to overall drier conditions during 'EOF1-'), and precipitation increases are confined towards southeastern Europe (Fig. 11, left panel). Meanwhile, over large regions north and west of the Alps, precipitation changes only weakly under these circulation conditions. Spatial patterns of dynamically-adjusted ensemble members (Fig. 11, middle panel) have a closer correspondence to the forced pattern than to the

spatial patterns of 'raw' 50-year trends (Fig. 11, right panel). Pattern RMSE is again substantially reduced (RMSE=0.052 for dynamically adjusted grid cells, RMSE=0.086 for raw trends).

Overall, we conclude that dynamical adjustment enables approximating the forced response from high resolution simulations with just a small ensemble of about three ensemble members. This is possible for both long-term trends in mean precipitation as well as for trends under more specific synoptic regimes. If only a small ensemble is available, RMSEs of 50-year precipitation

trends are reduced by up to about a factor of two (Fig. 8) by using dynamical adjustment based on the Latent Linear Adjustment Autoencoder.

## 5 Another potential application: Emulating dynamically-induced variability in daily precipitation fields

While we focus on dynamical adjustment, the proposed statistical model also lends itself to other applications. Here we describe how to leverage the Latent Linear Adjustment Autoencoder to obtain a weather generator for the dynamical component of the

climate model. Concretely, by bootstrapping (Efron, 1979) the SLP time series we obtain a new (bootstrap) sample $X^b$ on the basis of which we can generate predictions for the latent space variables $\hat{L}^b = h(X^b)$, which can then be decoded as $\hat{Y}_{X^b} = d(\hat{L}^b)$.

There are various options how to obtain the bootstrap sample of the SLP time series. Here, we present a simple first approach. Specifically, we use a simple block bootstrap for the SLP time series, i.e. we split the original SLP time series $X$ into non-

overlapping blocks (here, we choose a block length of 50 days). To create a new bootstrap sample $X^b$, we resample these blocks with replacement until the number of resampled days has reached the number of original days. By repeating this procedure $B$ times, we obtain different bootstrap samples $X^b$ for $b = 1, \ldots, B$. As an alternative to this simple block bootstrap, one could employ more elaborate bootstrapping techniques, such as a parametric bootstrap.

Resampling the low-dimensional SLP time series and using the Latent Linear Adjustment Autoencoder to produce the

corresponding precipitation fields has the advantage of avoiding the more complex and costly operations that depend on the high-dimensional spatial field for emulation. Modelling the spatial dependencies directly is challenging, while our technique leverages the trained models $h$ and $d$ to represent the relationship between the SLP time series and the dynamic precipitation component.





### 5.1 Demonstration of the Latent Linear Adjustment Autoencoder high-resolution precipitation emulator

A selection of example daily precipitation predictions (based on the original SLP time series $X$) and emulated predictions (based on different bootstrapped SLP time series $X^b$ for $b = 1, \ldots, 5$, all based on the holdout ensemble member "kbb") are shown in Fig. 12. The first column shows the precipitation predictions based on the SLP time series from the holdout ensemble member "kbb". The remaining columns show different emulated predictions $X^b$ for $b = 1, \ldots, 5$. As is to be expected, the emulated predictions based on the individual spatial fields are not visually distinguishable from the original predictions. Next,

we quantitatively assess the extent to which the precipitation distributions and temporal dependencies are preserved.

To quantify the difference between the emulated and RCM precipitation frequency and amount distributions (Pendergrass and Hartmann, 2014), we compute the Perkins' scores (Perkins et al., 2007). Note that the Perkins' score is a linear transformation of the Total Variation distance on the locally constant density defined by the histogram. Concretely, to compare the distributions of the predictions based on the coarsened RCM SLP time series to the predictions based on the bootstrapped SLP

time series, we pool the predictions from six holdout ensemble members ("kbb", "kct", "kcu", "kcv", "kcw", "kcx") and six emulated datasets (all based on the holdout ensemble member "kbb"). Pooling a number of timeseries has the advantage of reducing noise. We use the Perkins' scores to compare the precipitation frequency distributions for each grid cell. This skill score varies between 0 and 1, where 1 is perfect and 0 has no overlap; 0.9 is a very good skill score. The underlying histograms have evenly-spaced bins in log-space with a bin width of 0.4. The results are shown in Fig. 13. The Perkins' scores are very

high across all grid points, with little spatial variation. The domain mean is 0.983, with a standard deviation of 0.005. Hence, the emulated predictions maintain approximately the same frequency distribution of precipitation as the predictions based on the coarsened RCM SLP.

We furthermore compute the Perkins' scores based on the amount distribution of precipitation (Pendergrass and Hartmann, 2014). To calculate the amount distribution, we first sum the amount of precipitation in each bin of the precipitation histograms.

Subsequently, we normalize by the total amount of precipitation. Since the total amount of precipitation can differ between the predictions based on the coarsened RCM SLP time series and the emulations, we also show the relative difference between the total amounts of precipitation. In other words, the Perkins' scores only assess whether the two amount distributions have the same shape. The Perkins' scores for all grid cells are displayed in Fig. 14. Again, the Perkins' scores are very high throughout the domain. The domain average is 0.982 with a standard deviation of 0.010. Figure 15 shows the difference in

total precipitation in percent relative to the total RCM precipitation. Over land the variations are generally less than 5%, with the exception of some Alpine regions.

To examine the behavior of the emulated precipitation fields across timescales, we compare the power spectra of the predictions based on the original SLP time series and the emulated predictions. For each grid point and each holdout ensemble member/bootstrap sample, we compute the smoothed power spectrum and average the resulting curves. (The power spectrum

is computed using the R (R Core Team, 2020) function `spec.pgram` with a span of 20 of the Daniell smoothers.) Figure 16 shows the corresponding average power spectra. As can be seen, the distribution of power is well-preserved in the emulated





predictions. This shows that the coarse-resolution RCM SLP blocks enable maintenance of physically-plausible variations across timescales, which is a feature of the approach.

Because the bootstrapped samples depend entirely on the realizations that are available to draw from, we also test that the
emulated realizations are not overly similar to one another. Recall that the six emulated realizations we consider above are all based on the holdout ensemble member "kbb". Hence, we would like to assess whether the emulated data sets are more similar to the data from ensemble member "kbb", compared to other holdout ensemble members. For this analysis, Fig. 17 (left panel) shows the averaged cross-correlation functions between the precipitation predictions based on (a) six emulated SLP time series data sets and (b) the generating holdout ensemble member ("kbb") SLP time series, respectively. We evaluate the cross-
correlations at each grid cell and average over the whole spatial field. We contrast this averaged cross-correlation function with the cross-correlation functions of different holdout ensemble members ("kct", "kcu", "kcv", "kcw", "kcx") with the holdout ensemble member "kbb" in the right panel of Fig. 17. In this analysis, if the former cross-correlations (left panel) are not significantly larger than the cross-correlations in the latter comparison (right panel), it would imply that a set of emulated SLP realizations based on one holdout ensemble member are sufficiently diverse. Indeed, we observe that the cross-correlations in
the two comparisons are of the same order of magnitude and all are very small.

## 6 Conclusion and future work

In this work we have first introduced the Latent Linear Adjustment Autoencoder, which combines a linear model with the nonlinear decoder of a variational autoencoder. By combining a linear model, which takes a circulation proxy as input, with the expressive nonlinear (deep neural network) decoder, it can be easily trained, and allows for jointly modeling the dynamically-
induced high-resolution spatial field of the climate variable of interest. The main methodological novelty is that we add a linear model to the variational autoencoder and include an additional penalty term in the loss function that encourages linearity between the circulation proxy and the latent space. This leverages the advantages of a linear relationship between circulation variables and latent space variables, hence enhancing robustness, while also benefiting from the advantages of deep neural networks (i.e., flexibility in modeling non-linearities, such as those that occur in high-resolution orographic precipitation).
Future work targeting climate applications could explore adding other constraints or regularization to the linear model in the latent space, such as instrumental variables or anchor regression (Rothenhäusler et al., 2018) for distributional robustness (Meinshausen, 2018). Enforcing distributional robustness may be particularly interesting in the context of transfer learning, i.e., training the Latent Linear Adjustment Autoencoder on climate model simulations and applying the trained Latent Linear Adjustment Autoencoder to observations without the need for re-training.
Second, as the main application, we have tested the applicability of the Latent Linear Adjustment Autoencoder to dynamical adjustment of high-resolution precipitation based on daily data at regional scales. Based on a circulation proxy, the Latent Linear Adjustment Autoencoder predicts dynamic (circulation-induced) precipitation at high resolution. An estimate of the forced precipitation response can then be separated from internal variability, leaving higher signal-to-noise compared to raw multidecadal trends. With only a small ensemble of about 3 members, root mean squared errors are roughly halved compared to





raw trends when estimating the forced response (see Fig. 8), leading to dynamically-adjusted spatial trend patterns that closely resemble those of the approximated forced response (i.e., the ensemble average over 50 members), despite large internal variability. Moreover, we have used dynamical adjustment with the Latent Linear Adjustment Autoencoder to extend the framework to uncover estimates of the forced response conditioned on specific circulation regimes. We illustrated this aspect for composites of days with prevailing westerly conditions and hence wet conditions over Western Europe (similar to NAO+

regimes), and conversely, for days with suppressed westerlies (similar to NAO-) and hence generally drier conditions in Western Europe. In both cases the Latent Linear Adjustment Autoencoder was able to provide a better estimate of the forced response (i.e., with reduced error) compared to raw trends.

Other use cases of the Latent Linear Adjustment Autoencoder may include further applications of dynamical adjustment, including transfer learning across different high resolution simulations such as EURO-CORDEX models (Jacob et al., 2014).

Eventual application to observations for regional-scale detection and attribution of precipitation changes is anticipated. More general applications, such as statistical downscaling of the *circulation-induced* component of precipitation variability in coarse-scale GCMs, or in order to reconstruct observations at high resolution based on the prevailing large-scale circulation, are also conceivable. Importantly, the Latent Linear Adjustment Autoencoder requires only a coarse-scale circulation proxy (like SLP) to generate an estimate of dynamic precipitation at high resolution.

Lastly, beyond dynamical adjustment, we have illustrated how to leverage the Latent Linear Adjustment Autoencoder to emulate dynamically-induced variability in daily precipitation fields. While the results we have presented are promising in the sense that they produce realistic spatial and temporal patterns of daily precipitation over Central Europe, leveraging the relationship between large-scale circulation and precipitation, the technique should be adapted to and evaluated in more detail in the context of specific use cases.

Overall, the Latent Linear Adjustment Autoencoder may prove a versatile tool for climate and atmospheric science, specifically for modeling relationships between large-scale predictors and local and nonlinear precipitation at high resolution.

## Appendix A: Experimental details

In this section, we detail the architecture used for the encoder and decoder of the proposed model. Additionally, we report the most important hyperparameters. All further details can be found in the accompanying code; see the "Code and data

availability" section below for details. For the encoder and the decoder we use three convolutional layers and one residual layer (He et al., 2016) with filter sizes 16, 32 and 64 and a kernel size of 3. The dimensionality of the latent space $L$ is chosen to be 400. For the SLP time series we extract 750 components such that the linear model $h$ receives 750 predictor variables as input and has 400 target variables. The model is trained using the Adam optimizer (Kingma and Ba, 2015) for 100 epochs with a learning rate of $10^{-3}$. The penalty weight $\lambda$ in Eq. (2) is set to 1.





## Appendix B:  Experimental results for dynamical adjustment of daily temperature

The following results are based on temperature anomalies. In Fig. A1, we show examples from the holdout ensemble "kbb" of
(i) original temperature fields $Y$ (left column), (ii) reconstructions $\hat{Y}$ (center column), and (iii) predictions $\hat{Y}_X$ (right column).
The reconstruction quality, i.e. the similarity between the left and the center column, is quite high, even though not all fine
details are reproduced. For dynamical adjustment, we use the residuals, which are computed as the difference between the
original fields (left column) and the predictions $\hat{Y}_X$ (right column). The predictions are computed using the linear model $h$ and
the decoder $d$ with SLP time series as input. As can be seen, the proposed model yields spatially-coherent predictions and
explains a large proportion of the variance of $Y$.

Figure A2 shows the gridcell-wise MSE and Fig. A3 shows the $R^2$ statistics for the holdout ensemble "kbb".

*Code and data availability.*  The Latent Linear Adjustment Autoencoder model is free and open source. It is distributed under the MIT soft-
ware license which allows unrestricted use. The source code is available at the following GitHub repository: https://github.com/christinaheinze/
latent-linear-adjustment-autoencoders. A snapshot of the source code used to generate the results presented in this manuscript is available at
https://doi.org/10.5281/zenodo.3957494 (Heinze-Deml, 2020a). The pretrained models can be obtained from https://doi.org/10.5281/zenodo.
3950044 (Heinze-Deml, 2020c). Finally, the input data is available at https://doi.org/10.5281/zenodo.3949747 (Heinze-Deml, 2020b).

*Author contributions.*  CH and NM conceptualized Latent Linear Adjustment Autoencoders. CH, SS, and NM conceptualized the climate
applications with support from AP and FL. CH, SS and NM developed the methodology. CH did the formal analysis. CH and SS did the
investigation and visualization. CH and SS wrote the original draft. CH, SS, AP, and NM reviewed and edited it.

*Competing interests.*  The authors declare that they have no conflict of interest.

*Acknowledgements.*  We thank Raul Wood and Martin Leduc for providing the climate model simulations. The production of ClimEx was
funded within the ClimEx project by the Bavarian State Ministry for the Environment and Consumer Protection. The CRCM5 was developed
by the ESCER centre of Université du Québec à Montréal (UQAM; www.escer.uqam.ca) in collaboration with Environment and Climate
Change Canada. We acknowledge Environment and Climate Change Canada's Canadian Centre for Climate Modelling and Analysis for
executing and making available the CanESM2 Large Ensemble simulations used in this study, and the Canadian Sea Ice and Snow Evolution
Network for proposing the simulations. Computations with the CRCM5 for the ClimEx project were made on the SuperMUC supercomputer
at Leibniz Supercomputing Centre (LRZ) of the Bavarian Academy of Sciences and Humanities. The operation of this supercomputer is
funded via the Gauss Centre for Supercomputing (GCS) by the German Federal Ministry of Education and Research and the Bavarian State
Ministry of Education, Science and the Arts. This material is based in part upon work supported by the National Center for Atmospheric
Research, which is a major facility sponsored by the National Science Foundation (NSF) under Cooperative Agreement No. 1947282, and



by the Regional and Global Model Analysis (RGMA) component of the Earth and Environmental System Modeling Program of the U.S. Department of Energy's Office of Biological & Environmental Research (BER) via NSF IA 1844590. S.S acknowledges funding provided

by the Swiss Data Science Centre within the project 'Data Science-informed attribution of changes in the Hydrological cycle' (DASH, ID C17-01).



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



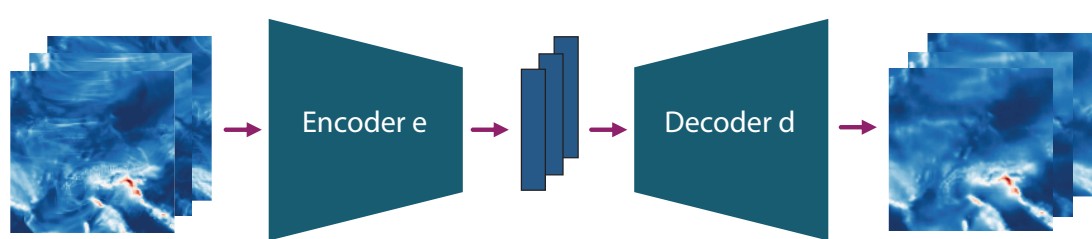

**Figure 1.** Illustration of a standard autoencoder model: The spatial fields $Y$ are fed to the encoder $e$ which maps them to the latent space variables $L$ (illustrated in blue). These are in turn fed to the decoder $d$ which computes a reconstruction of the input, $\hat{Y}$.





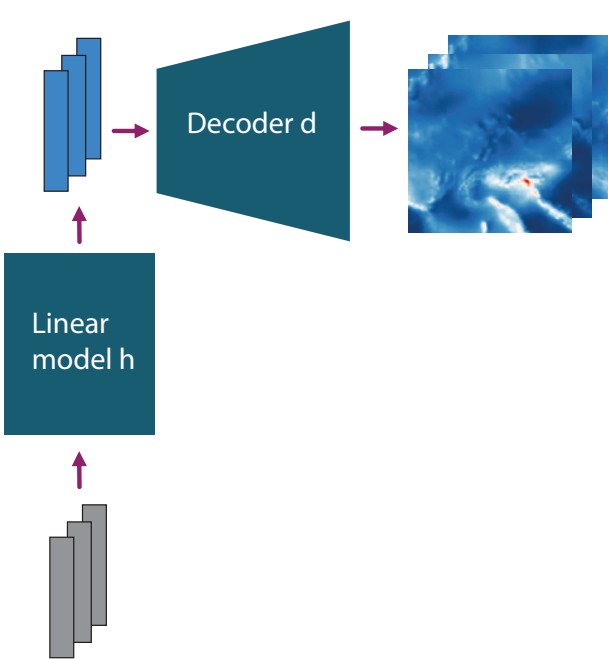

**Figure 2.** Illustration of the Latent Linear Adjustment Autoencoder *after training*: The input features $X$ (illustrated in gray) are fed to the linear model $h$ which yields a prediction for the latent space variables $L$ (illustrated in light blue). These are in turn fed to the decoder which computes a prediction of the spatial field based on $X$ only, $\hat{Y}_X$. Training the models $e$, $d$ and $h$ is performed iteratively in an alternating fashion.







**Figure 3.** Examples of (i) original precipitation fields (left column), (ii) reconstructions (center column), and (iii) predictions (right column). For better visibility, data are square-root-transformed. Hence, the units are $\sqrt{\mathrm{mm}} \cdot \mathrm{d}^{-1}$.

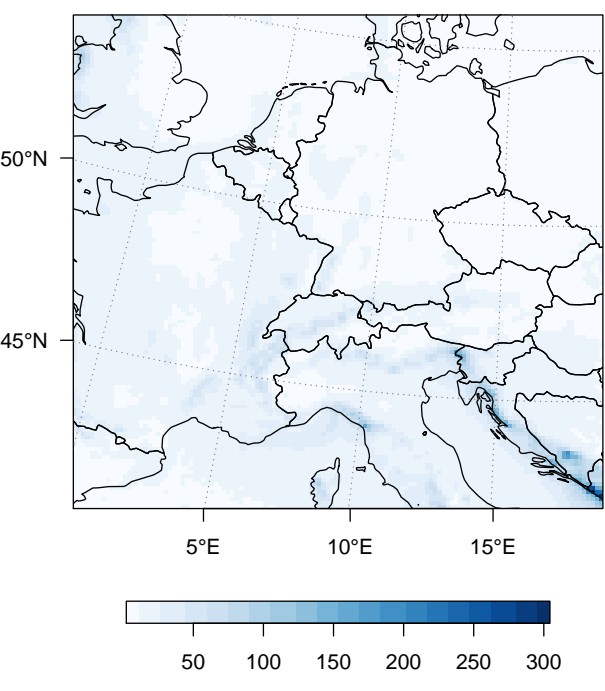

**Figure 4.** Mean-squared error (MSE, based on precipitation data in mm·d$^{-1}$) for each grid cell for the precipitation predictions.



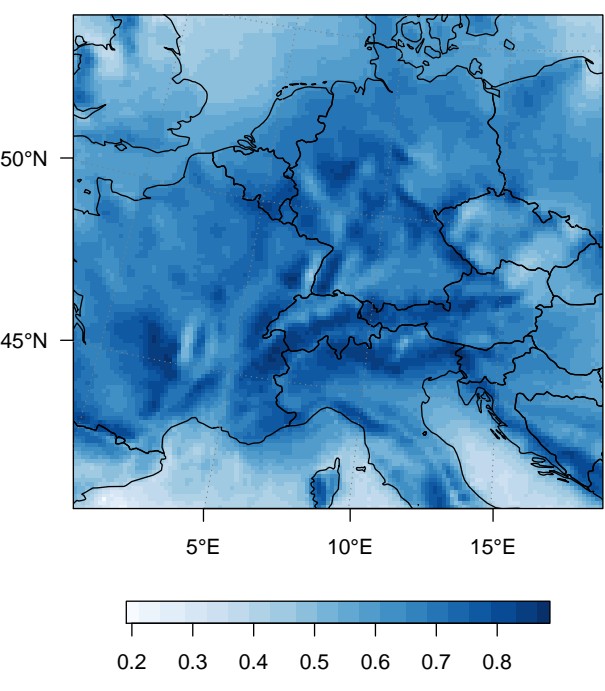

**Figure 5.** Proportion of variance explained ($R^2$) for each grid cell for the precipitation predictions.





**Figure 6.** Dynamical adjustment of the full domain (land grid cells only) using autoencoders. (left) seasonal mean precipitation simulated by three members of the high-resolution RCM in black (top: 'kbb', middle: 'kct', bottom: 'kcu'), the predicted ('circulation-induced') component for three 'holdout' ensemble members (blue) and the forced response (average across all 50 members, red). (right) Residuals from the prediction (black dots) and the forced response (red).







**Figure 7.** Dynamical adjustment of 50-year winter precipitation trends (2020-2069). (Left column) Forced precipitation response (2020-2069 linear winter precipitation trends averaged over 50 ensemble members). (Middle Column) Linear 50-year winter precipitation trends in three randomly selected dynamically adjusted ensemble members (top: 'kbb', middle: 'kct', bottom: 'kcu'). (Right column) Linear 50-year winter precipitation trends in the original ensemble members (top: 'kbb', middle: 'kct', bottom: 'kcu'). (Bottom row) Scatter plots of 50-year precipitation trends in the forced response against 50-year precipitation trends over land in dynamically adjusted (middle) and originally simulated ensemble members (right).





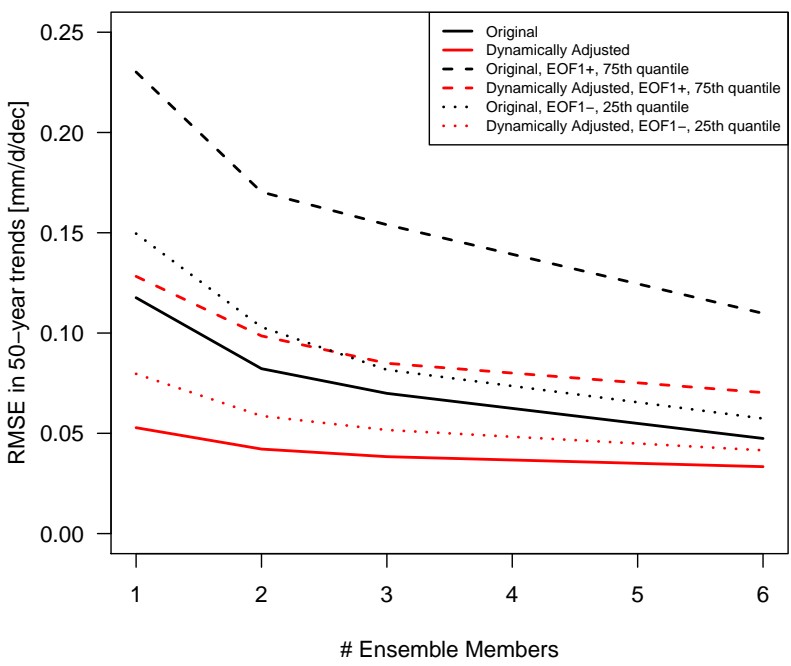

**Figure 8.** Reduction in RMSE as a function of the number of ensemble members used in dynamical adjustment.



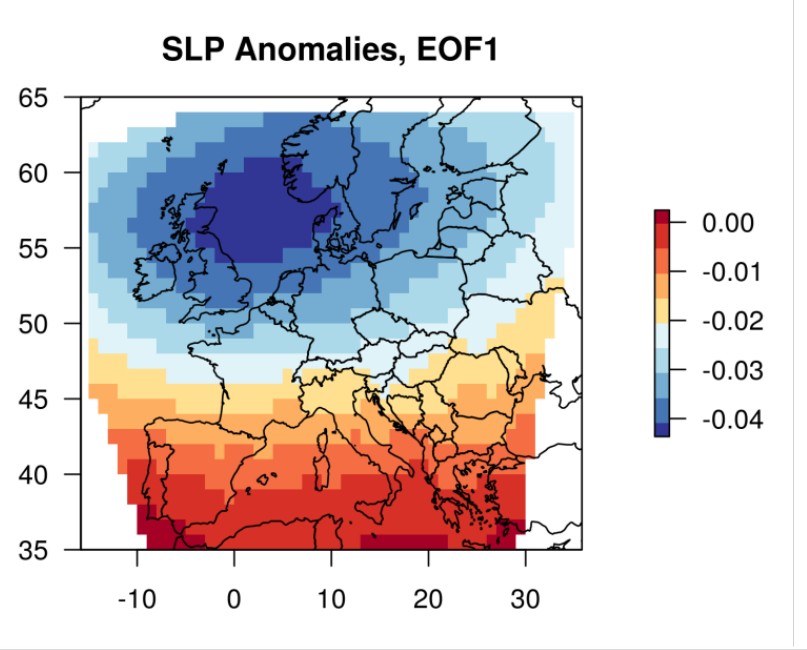

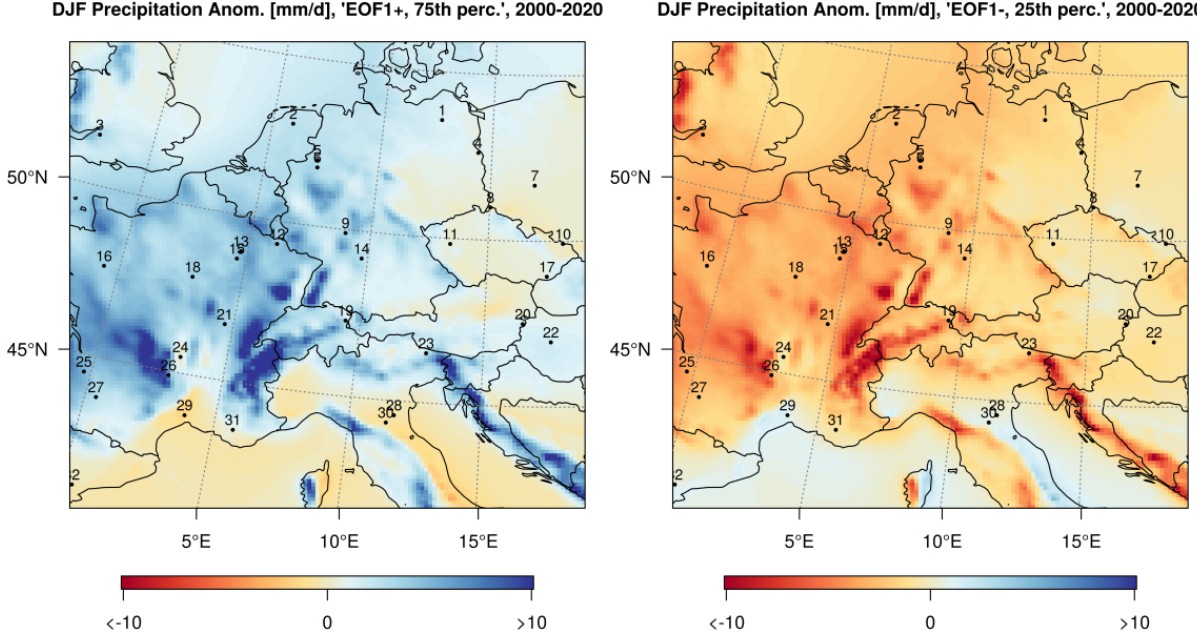

**Figure 9.** (top) First Empirical Orthogonal Function of Sea Level Pressure over the full domain of the regional model. (Bottom) Winter precipitation anomalies for the 25% of days that show the strongest (left) and weakest (right) projection on the pattern of the first EOF (i.e., days with a strong zonal SLP gradient and hence dominant westerly flow (left), and days with weak to absent zonal SLP gradients (right)).





**Figure 10.** Dynamical adjustment of 50-year winter precipitation trends (2020-2069) under the 'EOF1+' regime (25% of all days that project strongest on the first EOF, i.e., that show a strong westerly flow). (Left column) Linear forced winter precipitation trends under the 'EOF1+' regime (2020-2069). (Middle Column) Linear 50-year winter precipitation trends in an average of three randomly selected dynamically adjusted ensemble members. (Right column) Linear 50-year winter precipitation trends in an average of three corresponding original ensemble members. (Bottom row) Scatter plots of 50-year precipitation trends in the forced response against 50-year precipitation trends over land in dynamically adjusted (middle) and originally simulated ensemble members (right).





**Figure 11.** Dynamical adjustment of 50-year winter precipitation trends (2020-2069) under the 'EOF1-' regime (25% of all days that project weakest on the first EOF). (Left column) Linear forced winter precipitation trends under the 'EOF1-' regime (2020-2069). (Middle Column) Linear 50-year winter precipitation trends in an average of three randomly selected dynamically adjusted ensemble members. (Right column) Linear 50-year winter precipitation trends in an average of three corresponding original ensemble members. (Bottom row) Scatter plots of 50-year precipitation trends in the forced response against 50-year precipitation trends over land in dynamically adjusted (middle) and originally simulated ensemble members (right).





**Figure 12.** Examples of (i) predictions based on the original input data $X$ (leftmost column) and (ii) emulated predictions based on bootstrapped input data $X^b$ (remaining columns). Here, we show emulated examples from five different bootstrap samples $X^b, b = 1, \ldots, 5$. For better visibility of the precipitation features, the figure shows square-root-transformed data. Hence, the units are $\sqrt{\text{mm}} \cdot \text{d}^{-1}$.



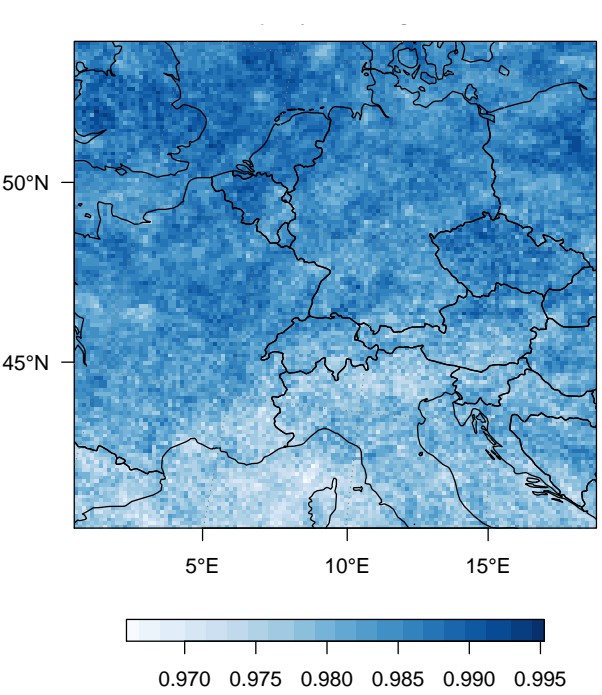

**Figure 13.** Perkins' scores for the frequency distribution





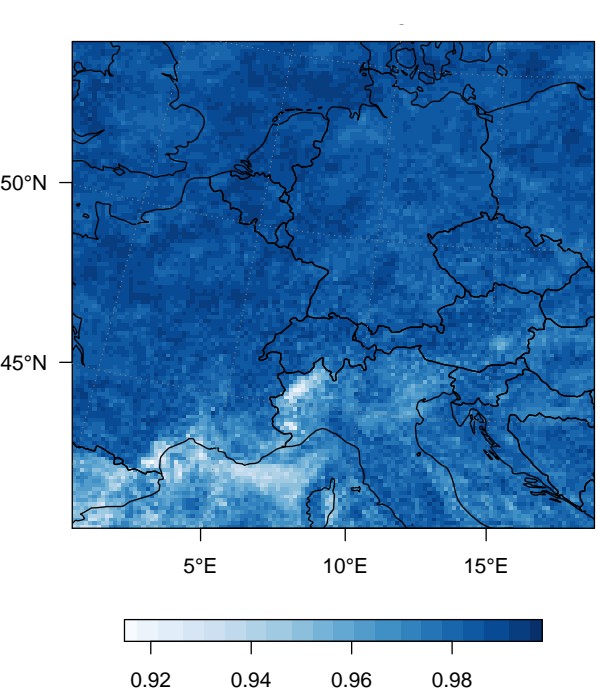

**Figure 14.** Perkins' scores for the amount distribution



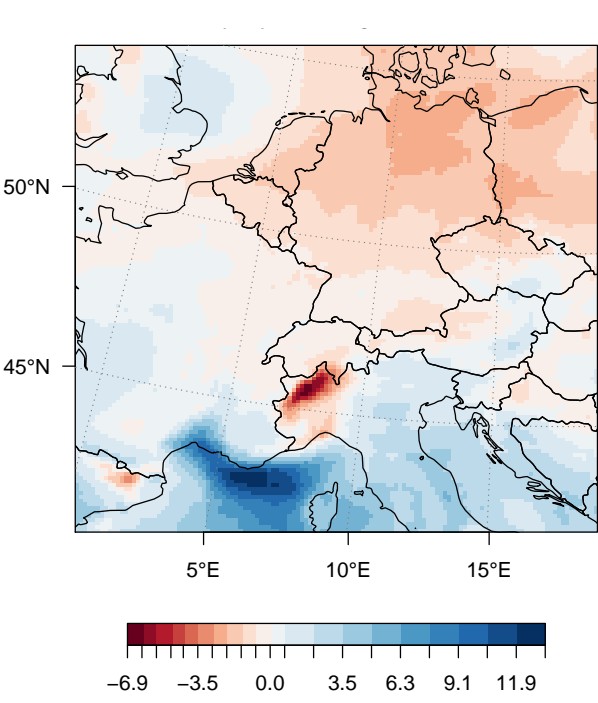

**Figure 15.** Relative difference in total precipitation between the original and the emulated predictions (in %)



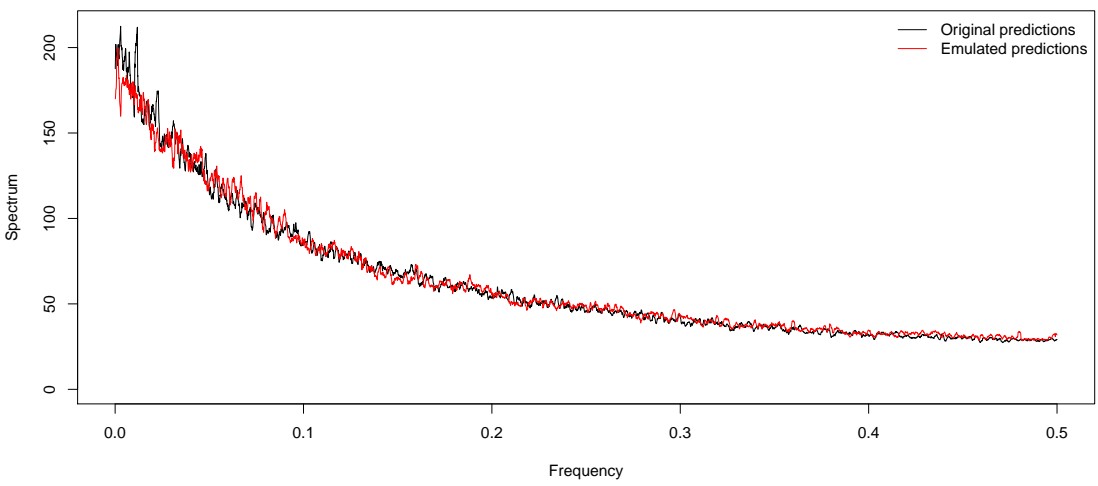

**Figure 16.** Average power spectrum of (i) the original predictions (black) and (ii) the emulated predictions (red). The power spectra are averaged over grid points and six samples (i.e., six different holdout ensemble members and six different bootstrapped data sets), respectively. Frequency is displayed in cycles per day and the spectrum is scaled such that the area under the curve equals the variance of the signal.





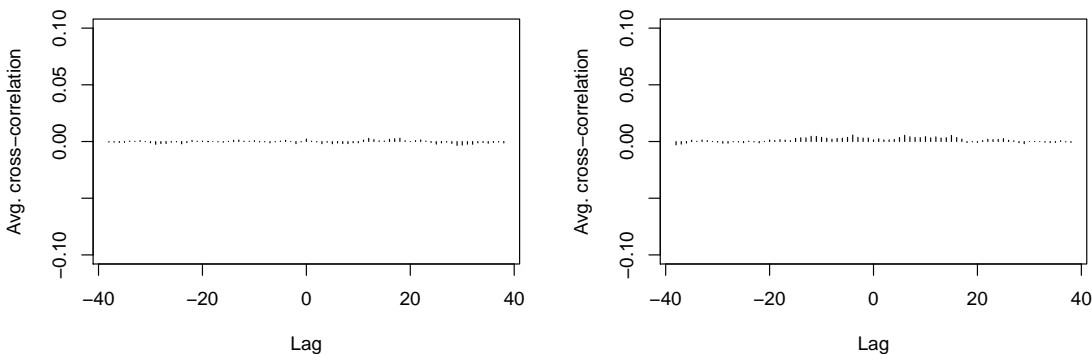

**Figure 17.** Left: Average cross-correlation functions between six different emulated data sets of dynamically-induced precipitation and the predicted dynamically-induced precipitation based on the original input data (from ensemble member "kbb"), respectively. The cross-correlation functions are computed at each grid cell and averaged over the whole spatial field. Right: Equivalent computation, contrasting the predicted dynamically-induced precipitation of five holdout members ("kct", "kcu", "kcv", "kcw", "kcx") with the holdout ensemble member "kbb". The cross-correlations in the right and left panel are of the same order of magnitude and all very small. The difference between the two cross-correlation functions is irrelevant due to the small magnitudes and can hence be disregarded.



**Figure A1.** *Temperature anomalies.* Examples of (i) original temperature fields $Y$ (left column), (ii) reconstructions $\hat{Y}$ (center column), and (iii) predictions $\hat{Y}_X$ (right column).





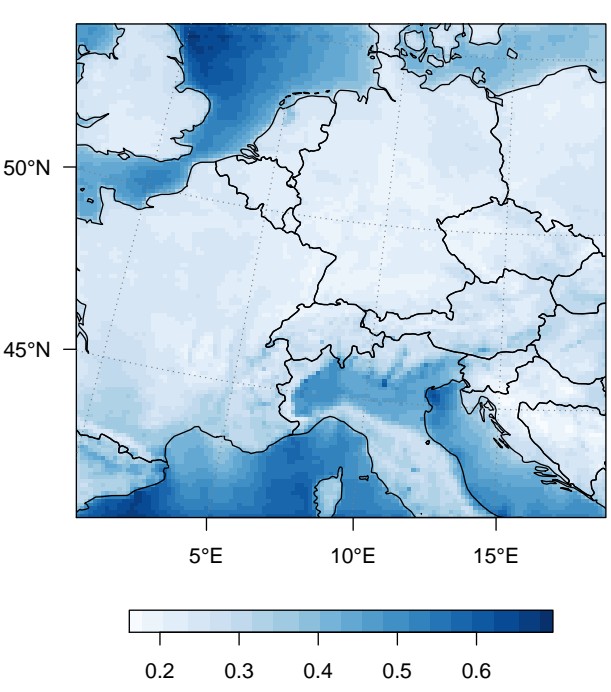

**Figure A2.** *Temperature anomalies.* Mean-squared error (MSE) for each grid cell for the temperature predictions.





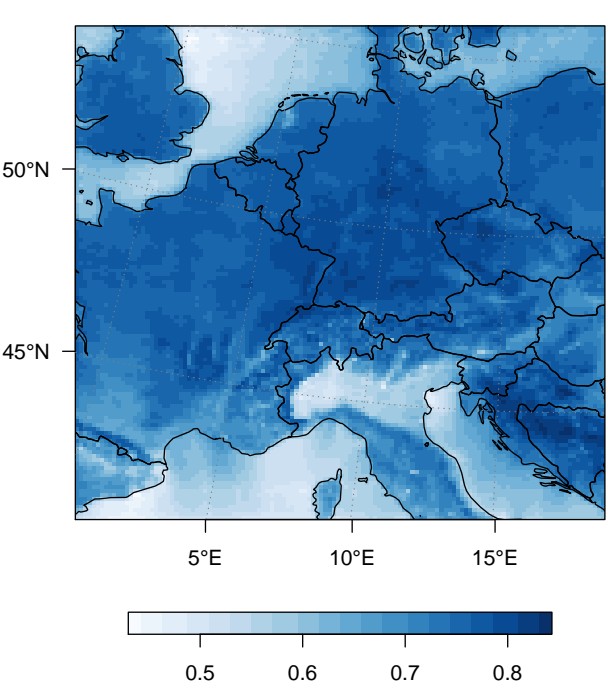

**Figure A3.** *Temperature anomalies.* Proportion of variance explained ($R^2$) for each grid cell for the temperature predictions.