# Peer review of "Latent Linear Adjustment Autoencoders v1.0: A novel method for estimating and emulating dynamic precipitation at high resolution"

_Geoscientific Model Development, 2020_

## Referee Comment (RC1) · Anonymous Referee #1 · 6 Jan 2021

This article is a novel and very interesting piece of work, with potential for applications in the field of climate science, some of which are presented in the article. The authors describe a latent adjustment autoencoder modified with the addition of a linear component between the input sea level pressure Empirical Orthogonal Function timeseries and the latent space of the autocoder. In a first application, the authors show that this allows to remove the internal variability of winter precipitation over Europe and extract the thermodynamical forced signal from only a few members of simulations instead of averaging a very large number of simulations. A second application they show is a weather generator, based on bootstrapping the SLP EOFs and then decoding the precipitation fields. All applications are limited to generating present-day like winter

precipitation patterns since the only input variable is sea level pressure. It may limit the application of the method to other seasons where precipitation may be less tied to SLP patterns. Although I am not a specialist in machine learning, I found the method well-explained and I had a look at the code which also seems well-explained and portable for the use by others. However, I would like some additional discussion in the text and a few changes to the figures before it can be accepted.

**1  Major comments:**

On the method itself:

1. I don't understand why you train your autocoder on 1955-2070 data. There is a chance that you include some thermodynamical signal in the precipitation field when you minimise $Y - \hat{Y}_X$. I understand that you detrend the SLP EOF time series, but you don't detrend precipitation. Why not training on 1955-1995 and potentially use more members to have the same amount of data?

2. It would be good to know the minimal amount of data needed to train the algorithm. Indeed, if 1955-2070 daily data from a 9 member ensemble is needed to train the algorithm, then it would be cheaper to directly calculate the forced response from this 9-member ensemble (see comments below on Fig. 8) without dynamical adjustment. Ideally, one would like to dynamically adjust expensive simulations which cannot be run for long periods of time (e.g. a few decades).

**2  On the examples of application:**

1. I am convinced by the use of the new tool for dynamical adjustment on a large domain and seasonal scale (Fig. 6), this seems to be very successful, even

with only 1 member. This is quite impressive. For more detailed spatial scales however, it is less successful and I guess from extrapolating Fig. 8 that using 7 or 8 members for the "traditional runs" (out of 50) outperforms the dynamical adjustment. I would like to see more discussion on this in the text and I think that Fig. 8 could be improved with a few changes:

- extend the x axis to at least 10 members, to see when a "traditional averaging" outperforms the dynamical adjustment (this implies performing dynamical adjustement on more holdout members).

- you plot only one value and it does not correspond to the one in the text (line 255), I presume for 1 member you can have 41 different values (excluding the training set), so you can add median + inter-quartile range / sqrt(number of samples), so that one knows if the difference is statistically significant, but I presume so.

- Add details to the caption. I presume that it shows the RMSE of 50y trend maps calculated by averaging n members compared to 50y trends using 50 member average. It is not very clear from the caption.

2. The tool is successful for seasonal means. Can you comment on the potential use of this tool for assessing trends in extreme precipitation, for which regional models are more trustworthy than global models? The prediction in precipitation fields seems smoothed out compared to original fields. And not taking into account thermodynamical fields as predictors may be limiting the representation of extremes, even in a present-day context.

3. Regarding the weather generator, I do struggle to exactly understand the novelty of your method. If I understand correctly, you are bootstrapping the time series of EOFS, but keeping each daily EOF set as it is, so you are not "creating" new pressure patterns, just shuffling them. One could do this directly by shuffling daily precipitation maps in the same way. I agree that one would need 150 years of

present-day data instead of simulations with evolving greenhouse gases, but this is easily achieved these days. It is interesting that you show that shuffling 150 years of data seems as good as running several members, at least for the bulk of precipitation distribution. I wouldn't think this is true for extremes. I think the use for dynamical adjustment has much more potential than the weather generator.

I would suggest to reduce this section to have more space in the article for a figure to reply to my point 2 about the method.

**3 Minor comments:**

Fig. 7, 10, 11: the scatter plots are saturated, it may be better to plot a gaussian kernel density estimation https://seaborn.pydata.org/generated/seaborn.kdeplot.html

Most figures with blue shading only: I find the continuous colour shading difficult, it may be best to reduce the number of colour levels used. One could also potentially use a sequential colour map like terrain_r for precipitation fields. It will make figures more readable and may reduce the need to show square root precipitation fields, which are less intuitive.

Fig. 9: remove the numbers on it, you are not using them in the article.

"As is to be expected, the emulated predictions based on the individual spatial fields are not visually distinguishable from the original predictions." Do you mean that they look "physical" with no artefacts? They are not meant to be similar to the original predictions. This is just like Fig. 3, I don't really see the point of this figure.

Fig. 12: caption could be a bit more wordy to be self-explanatory if readers only partially read the article.

Typos: Line 17: are expected to remains in the Line 176: Fig. 7a -> Fig. 9a

---

## Referee Comment (RC2) · Anonymous Referee #2 · 19 Feb 2021

This manuscript describes a novel and potentially very useful machine learning technique for separating forced signals and internal variability from climate model output (as well as other applications such as weather generation). I should say at the outset of this review that I am an expert on climate, not on machine learning, so I cannot comment on detail on the machine learning method in this study. From what I can determine though, the methodology looks mainly sound and produces sensible results. The manuscript is very well written and logically presented. I am recommending major revisions because I would like to see a sensitivity test on the time period of the training data, but other than that my comments are mainly relatively minor.

Major points:

1) The training (1955-2070) and testing (2071-2100) periods are consecutive, which I do not think is the best choice, as the training data is likely to contain a forced precipitation trend. Separating the training and testing datasets (e.g. training 1955-2020) would provide a more rigorous test of whether the dynamical adjustment method can separate internal variability from a forced signal, without much of the forced signal being present in the training dataset. The authors should test at least some of their results for sensitivity to the choice of training period.

2) I am not convinced that the forced signal that is extracted using the dynamical adjustment method is a purely thermodynamic signal of precipitation change, for two reasons. Firstly, the residual trend will include not only Clausius-Clapeyron-related increases in moisture, but also any other change in the relationship between SLP and precipitation under climate change. This could include, for example, changes in land-atmosphere interactions or weather system dynamics.

Secondly, there may be changes in the pattern of the individual SLP EOFs under climate change. Even small changes could have large consequences for regional precipitation. The authors have tried to address this point by detrending the SLP time-series, based on trends in EOF1, but I was slightly confused by the description of this detrending, and am not convinced that it would account for any (possibly subtle) changes in the shape of EOFs.

I think this is mainly a question of interpretation. The dynamical adjustment method will (as I understand it) remove any signal caused by temporal variation in the frequency of the SLP EOFs that were identified during the training period. The removed component will likely be due mainly to internal variability, though it could also include some forced signal if forcing were to drive any systematic change in the relative frequency of SLP EOFs. The residual will likely be a forced signal but I think calling it a thermodynamic precipitation change is too much of an oversimplification to be useful. Other factors

could also be important.

3) Dynamical adjustment appears to have the potential to significantly reduce the size of ensembles needed to reliably extract forced trends. However, a certain number of model years are needed to train the algorithm, so it is not clear exactly what the computational cost saving would be overall. Could the authors provide an estimate of the overall fractional saving in computational cost, taking algorithm training into account?

4) Is there an alternative type of machine learning algorithm that could be used to link SLP EOFs as input directly to the 2D precipitation fields as output (e.g. some form of neural network)? What are the benefits of using the intermediate stage of the autoencoder? I am not suggesting any extra analysis here, only for the authors to justify their choice of method a bit more.

Minor points: 1) It is not clear from the objectives in section 1 that the dynamical adjustment will be used to separate forced precipitation trends from internal variability. It would be useful to the reader for this objective to be spelled out here.

2) Fig. 3: How were these examples chosen? Are they representative of the data as a whole? It might be more useful to show high, medium and low skill cases rather than a random selection.

3) Figure colour scales. It is quite difficult to get much information out of the current single shading colour scales. I appreciate this is not a simple problem, but perhaps these could be improved to show the spatial features more clearly.

4) Fig. 4 & 5: Why only use a single holdout ensemble member for this? Why not use all of them? Also, relative error might be more informative for Fig. 4, rather than absolute error which mainly picks out the regions of high precipitation.

---

## Author Response (AR1)

**Final author response**

Submission "Latent Linear Adjustment Autoencoders v1.0: A novel method for estimating and emulating dynamic precipitation at high resolution"

We thank both referees for their insightful comments, helpful feedback and their positive evaluation. We address all points in detail below where we show the referees' comments in italics for ease of exposition.

**Please note:** In contrast to our previous response in the public discussion phase, we found one erroneous response in response to Reviewer #1's comments "On the example of applications": The number in the text (l. 259) in the original manuscript does indeed not match with Fig. 8, because Fig. 8 reflected the average across six holdout members, while the value in the text (taken from Fig. 7) reflects the RMSE across the three specific holdout members illustrated in Fig. 7; **both values reflect RMSE's over 50-year precipitation trends calculated for local grid cells**. We apologise for our previous confusing reply in the public discussion regarding this comment; please find more detailed comments and all changes made to Fig. 8 described in detail below.

**Review 1**

**Major comments:**

1.  *I don't understand why you train your autocoder on 1955-2070 data. There is a chance that you include some thermodynamical signal in the precipitation field when you minimise $Y − \hat{Y}x$. I understand that you detrend the SLP EOF time series, but you don't detrend precipitation. Why not training on 1955-1995 and potentially use more members to have the same amount of data?*

Thank you for raising these important points. We agree that there is a chance to include some thermodynamical signal under a long training period (although SLP is detrended). We also agree that it is important for applications to understand the sensitivity to (i) the training period choice, (ii) the amount of training data as well as (iii) the sensitivity to different detrending approaches. Hence, we perform the following additional analyses:

(1) **Sensitivity test to training data amount and training period**:

We train on a shorter period from 1955-2020 (as suggested by referee 2), using the same ensemble members as in the first submission (i.e., equivalent to a ~43% reduction in the training data, but more importantly, restricting the training to a period with relatively modest precipitation change). We then reproduce the dynamical adjustment analysis with this model

trained on (i) this shorter time period; and (ii) less data. We find that all performance measures (i.e., mean squared error, etc.) indicate very robust results with respect to these changes in the input data. In particular, the dynamical adjustment analysis based on the shorter training period reveals almost identical results as compared to the longer 1955-2070 training period. That is, the residual variability is much closer to the ensemble mean forced response (see plot reproduced below: Fig. 1, compare to Fig. 6 in the main manuscript). This sensitivity analysis thus provides support that our method is robust to (i) a shorter time period and (ii) less training data points. The detailed results can be found in Appendix B of the revised manuscript (Figs. A1-A3).

[Figure]

Figure 1: Dynamical adjustment analysis for the Latent Linear Adjustment Autoencoder (LLAAE) model trained only on the period 1955-2020 and thus with 43% fewer training data points. Compared to Fig. 6 in the main paper, which shows the same analysis for the LLAAE model with more training data (1955-2070), the results shown here are almost identical.

(2) **Trend removal sensitivity test**:

As correctly pointed out by the reviewer, the question of whether and how to detrend prior to dynamical adjustment is open, somewhat subjective, and often discussed as an inherent subjective choice/uncertainty in dynamical adjustment papers (see, e.g. Deser et al. 2016, or Lehner et al. 2017, and Lehner et 2018, for a discussion about trend removal). We agree that more discussion on this point is needed in the revised manuscript, and we have added this discussion in lines 320-335.

Forced changes in European winter SLP are highly uncertain, and models disagree on the sign and patterns of forced circulation change (Fereday et al. 2018) - although a northward shift in storm tracks and a dynamical extension of the subtropical dry zones is generally expected but not supported by all models (Fereday et al. 2018). Thermodynamic aspects are typically considered more robust across models (Shepherd et al. 2014; Fereday et al. 2018).

As pointed out by the reviewer, we have orthogonalized SLP EOF time series w.r.t. the **ensemble-mean SLP change** over time (i.e., a very simplistic but generic "detrending"). Our main motivation to use this somewhat simplistic detrending approach in our proof-of-concept study was to avoid that the ML method would take hypothetical dynamical changes to predict thermodynamical trends in precipitation (as the reviewer correctly pointed out). In other words, our goal was to estimate daily precipitation variability at high resolution *in absence of SLP* changes; that is introducing LLAAEs as a versatile tool for estimating a high-resolution precipitation field based on a coarse-resolution sea level pressure field.

Our analysis shows that the residuals match the ensemble mean very well (Fig. 6 in the main manuscript). Hence, if a trend signal would be included in the prediction of the precipitation field (e.g., due to hypothetical remaining trend artefacts in the pressure field), this effect is likely to be small because the residuals match the ensemble mean (forced) trend very well.

However, in addition to the results so far, we test an alternative simple detrending approach, where SLP is not detrended, but where we detrend precipitation using a simple LOESS smoother, fitted on the ensemble (seasonal) means at every location and subtracted from every day individually. (Furthermore, we here use the shorter 1955-2020 period for training the model.) For the dynamical adjustment analysis, we then compute the residuals based on the non-detrended precipitation data and our predictions (from the model trained on the detrended precipitation data; see Fig. 2). This analysis suggests that this approach to detrend precipitation is too simplistic since the residuals of the dynamical adjustment analysis underestimate forced changes (the ensemble mean) to some extent (Fig. 2). There are several possible reasons for this:
(1) Precipitation change cannot be modelled by a single additive mean change across the whole distribution. For instance, precipitation change is known to increase the variance of the precipitation distribution (Pendergrass et al. 2017). Hence, by subtracting the estimated, seasonally averaged precipitation trend we may have not fully removed the trend for wet days.

Developing a more refined approach to remove the forced precipitation changes from daily data is non-trivial and beyond the scope of this work, but will be addressed in future work.
(2) There may be some dynamically-induced changes in precipitation, but it would be difficult to evaluate this without any additional simulations where dynamical effects and thermodynamical effects could be separated.

Overall, we conclude that our simple SLP detrending (without detrending precipitation) is a useful approach for introducing LLAAEs as a versatile tool for dynamical adjustment, as demonstrated by the fact that the residuals of individual ensemble members after dynamical adjustment match the ensemble mean trend of precipitation very well (e.g., Fig. 6 in the main manuscript). However, we acknowledge that considerations around whether and how to detrend the data prior to dynamical adjustment are crucial, especially for real-world applications. We discuss this in the revised manuscript (lines 320-335 and Appendix B), and we acknowledge that more work is needed to fully understand the effect of detrending choices, which however is beyond the scope of the present study.

[Figure]

Figure 2: Dynamical adjustment analysis for LLAAE model trained only on 1955-2020 period and with detrending only precipitation using a LOESS smoother.

2. *It would be good to know the minimal amount of data needed to train the algorithm. Indeed, if 1955-2070 daily data from a 9 member ensemble is needed to train the algorithm, then it would be cheaper to directly calculate the forced response from this 9-member ensemble (see comments below on Fig. 8) without dynamical adjustment. Ideally, one would like to dynamically adjust expensive simulations which cannot be run for long periods of time (e.g. a few decades).*

We provide some analysis in that direction by limiting the training period to 1955-2020 (see above). This reduction in training data does not have a noticeable effect on the performance of the model (see above). In general, "the minimal amount of data" will depend on one's requirements of how to use the method. We expect the performance of the method to decrease gradually when further reducing the amount of training data.

We agree with the reviewer that, as machine learning algorithms are known to require rather large amounts of training data, "proving" the usefulness of autoencoder dynamical adjustment in a large ensemble may not be as straightforward (as correctly pointed out by the reviewer, we used nine ensemble members for training, which we could have used instead for calculating a 9-member ensemble average). However, we anticipate the ultimate applications of autoencoder-based dynamical adjustment not on a large ensemble (where the forced response is "known" anyways, to some extent), but instead on simulations with models where only one (or very few) ensemble members may be available. Hence, our present manuscript is intended as a proof-of-concept of the method within a large ensemble. As the next step, we envision the application to different climate models (e.g. training on a large ensemble or multiple large ensembles, and application of the dynamical adjustment to models for which only a few simulations exist), and with ultimate application of the trained autoencoders on reanalysis SLP data. This would allow us to leverage the available data from climate model simulations while applying the method in a context where a direct calculation of a multi-member ensemble mean is not possible. We discuss and clarify this point in the revised manuscript in lines 336-358 and Appendix B.

**On the examples of application:**

1. *I am convinced by the use of the new tool for dynamical adjustment on a large domain and seasonal scale (Fig. 6), this seems to be very successful, even with only 1 member. This is quite impressive. For more detailed spatial scales however, it is less successful and I guess from extrapolating Fig. 8 that using 7 or 8 members for the "traditional runs" (out of 50) outperforms the dynamical adjustment. I would like to see more discussion on this in the text and I think that Fig. 8 could be improved with a few changes:*

- *extend the x axis to at least 10 members, to see when a "traditional averag- ing" outperforms the dynamical adjustment (this implies performing dynamical adjustment on more holdout members).*
- *you plot only one value and it does not correspond to the one in the text (line 255), I presume for 1 member you can have 41 different values (excluding the training set), so you can add median + inter-quartile range / sqrt(number of*

*samples), so that one knows if the difference is statistically significant, but I presume so.*

Thank you for these suggestions, and for pointing out that Fig. 8 was thus far a bit unclear. We have improved Fig. 8 in the revised manuscript as suggested (figure is reprinted below/aside for ease of reading), by extending to 20 ensemble members and by bootstrapping from all 41 holdout members in order to show the full distribution and hence to better describe the variability/uncertainty of the reconstruction errors. We show the full distribution of RMSEs from 41 holdout members in the form of boxplots:

[Figure]

Furthermore, we have added additional discussion about the results shown in Fig. 8, as suggested by the reviewer (l. 261-272 in revised manuscript):

*"Fig. 8 (top panel) shows the RMSE for the reconstruction of forced 50-year precipitation trends (i.e., the 50-member average), via dynamical adjustment and the averaging of original ensemble members, as a function of the number of ensemble members n. With an increasing number of ensemble members (n), the reconstruction RMSE of the forced response reduces considerably. Hence, dynamical adjustment is particularly useful when only few members are available; e.g. for small ensembles up to five members. If only one member is available, the reconstruction RMSE of the forced 50-year precipitation trend is reduced by more than half via dynamical adjustment. Conversely, to achieve the same RMSE of a single dynamically adjusted ensemble member, an ensemble average of about four to six members would be required (Fig. 8, top panel). On the other hand, for ensembles with more than about 14 members, dynamical adjustment does not improve the ability to reconstruct the forced response. Moreover, dynamical adjustment reduces not only the reconstruction RMSE, but also reduces the spread*

[Figure]

[Figure]

*of the distribution across ensemble members, as indicated by the boxes and whiskers in Fig. 8 (top panel). The overall reduction of the reconstruction RMSE also holds particularly for specific circulation regimes (Fig. 8, middle and bottom panel), and is discussed in the next subsection."*

Please note that the number in the text (l. 259) in the original manuscript did indeed not match with the figure, because Fig. 8 reflected the average across six holdout members, while the value in the text (taken from Fig. 7) reflects the RMSE across the three specific holdout members illustrated in Fig. 7. However, please note that in the revised manuscript the RMSE values quoted in the text (l. 259, identical values to original manuscript) fall well inside the distribution of RMSEs before and after dynamical adjustment as illustrated in Fig. 8.

Also, we apologize for our previous confusing comment in the public discussion regarding the reviewer's comment; now we believe all issues are fixed.

- *Add details to the caption. I presume that it shows the RMSE of 50y trend maps calculated by averaging n members compared to 50y trends using 50 member average. It is not very clear from the caption.*

Yes, exactly. We have improved the caption in the revised manuscript such that it reflects all details about the figure content: "RMSE of 50-year trends, calculated by averaging *n* members, compared to 50-year trends using 50-member ensemble average ('forced response'). RMSE's are based on land grid cells only and shown for averaging *n* original ensemble members (black) and averaging *n* dynamically adjusted ensemble members (red). Trends are calculated over the entire DJF season (top), and only for EOF1+ (middle) and EOF1- situations (bottom). Boxplot whiskers indicate 2.5th and 97.5th percentiles (boxes show 25th and 75th percentiles) of RMSE distribution obtained from bootstrapping from the 41 holdout ensemble members".

2. *The tool is successful for seasonal means. Can you comment on the potential use of this tool for assessing trends in extreme precipitation, for which regional models are more trustworthy than global models? The prediction in precipitation fields seems smoothed out compared to original fields. And not taking into account thermodynamical fields as predictors may be limiting the representation of extremes, even in a present-day context.*

Extreme precipitation is important and there is a demand for information about these events at as high spatial resolution as possible. Our autoencoder may be able to fill an important gap in constructing extreme events in that it can reconstruct the dynamical component of extreme precipitation events (at least, the component proportional to surface pressure). Estimating the thermodynamic component is generally more straightforward than the dynamical component, and it may be possible to estimate it with other more straightforward approaches, particularly in winter. The reviewer is correct though, that the autoencoders, similar to other statistical/ML techniques, have a tendency to "smooth out" predictions (and thus probably underpredict the most extreme precipitation days). However, the technique may still be an improvement over existing alternatives: the resulting smoothing may still be less than the effective smoothing that

occurs at the coarse resolution climate models. Rigorous evaluation of this application is, however, beyond the scope of this manuscript.
We have added discussion around extreme events in the section that discusses the application of LLAAEs to composites of specific circulation regimes (lines 306-311):

*"The application of dynamical adjustment to composites of specific circulation regimes raises the question as to whether the Latent Linear Adjustment Autoencoder may be applicable to understanding the dynamical component in extreme precipitation events. While the LLAAE may be able to fill an important gap in reconstructing the dynamical component of daily precipitation fields, possibly including days with extreme precipitation (at least, the component proportional to surface pressure), it exhibits a tendency to smooth predicted precipitation fields (Fig. 3), which would presumably result in somewhat underpredicted extreme events. However, a detailed evaluation of the LLAAE in the context of extreme events will be the focus of future work."*

3. *Regarding the weather generator, I do struggle to exactly understand the novelty of your method. If I understand correctly, you are bootstrapping the time series of EOFS, but keeping each daily EOF set as it is, so you are not "creating" new pressure patterns, just shuffling them. One could do this directly by shuffling daily precipitation maps in the same way. I agree that one would need 150 years of present-day data instead of simulations with evolving greenhouse gases, but this is easily achieved these days. It is interesting that you show that shuffling 150 years of data seems as good as running several members, at least for the bulk of precipitation distribution. I wouldn't think this is true for extremes. I think the use for dynamical adjustment has much more potential than the weather generator.*

   *I would suggest to reduce this section to have more space in the article for a figure to reply to my point 2 about the method.*

Thank you for raising this concern. Since both referees suggested shortening this section, we have decided to follow this advice. Hence, we remove the section on the weather generator from our manuscript in order to focus on the method introduction and dynamical adjustment illustration, but we mention/discuss the possibility of weather generators here and we will expand on it elsewhere.

To answer your questions, note that the emulator generates dynamically-induced variability in the daily precipitation fields only. Hence, this cannot be achieved by using daily precipitation fields directly (additionally, note that we draw a bootstrap sample which is not equivalent to shuffling the data points).

Lastly, we agree that the performance for extremes is likely to be worse.

**Minor comments:**

*Fig. 7, 10, 11: the scatter plots are saturated, it may be better to plot a gaussian kernel density estimation https://seaborn.pydata.org/generated/seaborn.kdeplot.html*

We have changed the scatter plots to contour lines that show the 99% contours and 50% contours for each of the three illustrative ensemble members in Figs. 7, 10 and 11.

*Most figures with blue shading only: I find the continuous colour shading difficult, it may be best to reduce the number of colour levels used. One could also potentially use a sequential colour map like terrain_r for precipitation fields. It will make figures more readable and may reduce the need to show square root precipitation fields, which are less intuitive.*

Thank you for the suggestion. We have experimented with different color maps and different numbers of color levels but have not found alternative settings that yielded better figures.

*Fig. 9: remove the numbers on it, you are not using them in the article.*

We have removed the numbers.

*"As is to be expected, the emulated predictions based on the individual spatial fields are not visually distinguishable from the original predictions." Do you mean that they look "physical" with no artefacts? They are not meant to be similar to the original predictions. This is just like Fig. 3, I don't really see the point of this figure.*

*Fig. 12: caption could be a bit more wordy to be self-explanatory if readers only partially read the article.*

As we have decided to remove the section on the weather generator, we have also removed Fig. 12.

*Typos: Line 17: are expected to remains in the Line 176: Fig. 7a -> Fig. 9a*

Thank you, we have fixed the typos.

**Review 2**

**Major comments:**

1. *The training (1955-2070) and testing (2071-2100) periods are consecutive, which I do not think is the best choice, as the training data is likely to contain a forced precipitation trend. Separating the training and testing datasets (e.g. training 1955-2020) would provide a more rigorous test of whether the dynamical adjustment method can separate internal variability from a forced signal, without much of the forced signal being present in the training dataset. The authors should test at least some of their results for sensitivity to the choice of training period.*

Thank you, we provide two additional analyses: (1) to identify the **sensitivity to the training period**, and (2) to identify the **sensitivity to the trend removal procedure**. In short, reducing the training period to 1955-2020 does not have a noticeable effect on the results (see detailed discussion on this in the revised manuscript in Subsection 4.4, lines 336-346, and in Appendix B). The SLP trend removal procedure used in the paper is shown to yield residuals that match the ensemble mean precipitation trend very well. However, there is a known sensitivity of dynamical adjustment to different detrending choices, and we show this for the case when SLP would not be detrended but precipitation would be detrended (in a simplistic manner). We discuss the implication of this limitation of dynamical adjustment (generally) in the revised manuscript (Subsection 4.4, lines 320-335, and Appendix B). Please see our response to Major Comment 1 from the first referee for all details on both additional sensitivity tests.

2. *I am not convinced that the forced signal that is extracted using the dynamical adjustment method is a purely thermodynamic signal of precipitation change, for two reasons. Firstly, the residual trend will include not only Clausius-Clapeyron-related increases in moisture, but also any other change in the relationship between SLP and precipitation under climate change. This could include, for example, changes in land-atmosphere interactions or weather system dynamics.*

   *Secondly, there may be changes in the pattern of the individual SLP EOFs under climate change. Even small changes could have large consequences for regional precipitation. The authors have tried to address this point by detrending the SLP time-series, based on trends in EOF1, but I was slightly confused by the description of this detrending, and am not convinced that it would account for any (possibly subtle) changes in the shape of EOFs.*

   *The dynamical adjustment method will remove any signal caused by temporal variation in the frequency of the SLP EOFs that were identified during the training period. The removed component will likely be due mainly to internal variability, though it could also include some forced signal if forcing were to drive any systematic change in the relative frequency of SLP EOFs. The residual will likely reflect a forced signal but calling it a*

*thermodynamic precipitation change is too much of an oversimplification to be useful. Other factors could also be important.*

The reviewer raises very important points, and we agree. Referring to the residual as a pure "thermodynamic signal" is clearly an oversimplification. What we meant to say, and this is is agreement with the use of terminology in many dynamical adjustment papers (e.g. Deser et al 2016, Lehner et al 2017), is that we expect the residual time series to *contain* the imprint of thermodynamical signals, in particular thermodynamical changes (for example, as pointed out by the reviewer, increases in temperature that induce higher water-holding capacity of the atmosphere via the C-C relation). We have rephrased the revised manuscript such that, (1) it becomes very clear that we are not claiming that the residual is a purely thermodynamical signal, but that it may contain effects of feedbacks, remaining internal variability, circulation components not directly captured by SLP (lines 192-197), etc. In addition, long-term dynamical changes may even be part of the residuals; (2) we have also clarified that the choice of detrending (i.e., which variables to detrend, etc.) remains a key uncertainty in dynamical adjustment. Hence, clearly, more work is needed to fully understand the different implications of trend removal (see discussion in Subsection 4.4, lines 320-335, and Appendix B), but we believe this work is beyond the present study as the goal of this study was to illustrate Latent Linear Adjustment Autoencoders as a versatile tool to simulate daily precipitation variability based on a coarse SLP field.

3. *Dynamical adjustment appears to have the potential to significantly reduce the size of ensembles needed to reliably extract forced trends. However, a certain number of model years are needed to train the algorithm, so it is not clear exactly what the computational cost saving would be overall. Could the authors provide an estimate of the overall fractional saving in computational cost, taking algorithm training into account?*

Please see our reply to Point 2 made by Reviewer 1 for a more in-depth discussion. In short, we agree, making an exact calculation of "number of ensemble members saved" is difficult because of the training. But this is also not our main point: Our main point was to provide a proof-of-concept for applying dynamical adjustment to high-resolution regional precipitation fields (which is novel), and we anticipate eventually our model to be trained on large ensembles, but to be applied to, e.g. models where only few runs are available, or even reanalysis. We have included this discussion in Subsection 4.4 (lines 347-358).

4. *Is there an alternative type of machine learning algorithm that could be used to link SLP EOFs as input directly to the 2D precipitation fields as output (e.g. some form of neural network)? What are the benefits of using the intermediate stage of the autoencoder? I am not suggesting any extra analysis here, only for the authors to justify their choice of method a bit more.*

Linking SLP EOFs as input with the 2D precipitation fields as output without having the intermediate stage of the autoencoder would constitute a challenging estimation problem. Here, the autoencoder helps to estimate the decoder. We are not aware of alternative ML algorithms for this input/output combination and our methodology is novel in this regard.

More generally, one could extend the method of Sippel et al. by using a neural network instead of regularized linear regression. In that case, however, one would have a separate fit for each grid point (i.e. not the 2D precipitation field as output). This would be computationally demanding and it is also questionable whether the resulting predicted spatial field would be as coherent as what we obtained here. We have added this discussion in Subsection 4.5 "Alternative statistical and machine learning approaches".

**Minor points**

1. *It is not clear from the objectives in section 1 that the dynamical adjustment will be used to separate forced precipitation trends from internal variability. It would be useful to the reader for this objective to be spelled out here.*

Thank you. We have clarified this in the revised manuscript.

2. *Fig. 3: How were these examples chosen? Are they representative of the data as a whole? It might be more useful to show high, medium and low skill cases rather than a random selection.*

Thanks for the suggestion. The examples were chosen randomly but we have now updated the figure (Fig. 3 in the revised manuscript) to show examples for different quantiles of the loss (high, medium and low skill cases).

3. *Figure colour scales. It is quite difficult to get much information out of the current single shading colour scales. I appreciate this is not a simple problem, but perhaps these could be improved to show the spatial features more clearly.*

Thank you for the suggestion. We have experimented with different color maps and different numbers of color levels but have not found alternative settings that yielded better figures.

4. *Fig. 4 & 5: Why only use a single holdout ensemble member for this? Why not use all of them? Also, relative error might be more informative for Fig. 4, rather than absolute error which mainly picks out the regions of high precipitation.*

Fig. 4 and 5 look fairly similar for the other holdout members.
Regarding the relative error, we show the $R^2$ values in Fig. 3: Note that the $R^2$ values are computed as 1 - relative error where relative error = mean(residual sum of squares)/mean(total sum of squares).

Please also note that we have updated Fig. 8 to include all holdout ensemble members; and improved the characterisation of the distribution of RMSE's using a bootstrapping approach and all holdout members.

**References**

Deser, C., Terray, L. and Phillips, A.S., 2016. Forced and internal components of winter air temperature trends over North America during the past 50 years: Mechanisms and implications. *Journal of Climate*, 29(6), pp.2237-2258.

Fereday, D., Chadwick, R., Knight, J. and Scaife, A.A., 2018. Atmospheric dynamics is the largest source of uncertainty in future winter European rainfall. *Journal of Climate*, 31(3), pp.963-977.

Lehner, F., Deser, C. and Terray, L., 2017. Toward a new estimate of "time of emergence" of anthropogenic warming: Insights from dynamical adjustment and a large initial-condition model ensemble. *Journal of Climate*, 30(19), pp.7739-7756.

Lehner, F., Deser, C., Simpson, I.R. and Terray, L., 2018. Attributing the US Southwest's recent shift into drier conditions. *Geophysical Research Letters*, 45(12), pp.6251-6261.

Pendergrass, A.G., Knutti, R., Lehner, F. et al. Precipitation variability increases in a warmer climate. *Sci Rep* 7, 17966 (2017). https://doi.org/10.1038/s41598-017-17966-y

Shepherd, T.G., 2014. Atmospheric circulation as a source of uncertainty in climate change projections. *Nature Geoscience*, 7(10), pp.703-708

Sippel, S., Meinshausen, N., Merrifield, A., Lehner, F., Pendergrass, A. G., Fischer, E. M., and Knutti, R. (2019) Uncovering the forced climate response from a single ensemble member using statistical learning, *Journal of Climate*, 32, 5677-5699. doi:10.1175/JCLI-D-18-0882.1